# Metallogenic Mechanism of Typical Carbonate-Hosted Uranium Deposits in Guizhou (China)

**Lin-Fei Qiu [1,\*], Yu Wu [1], Qiong Wang [2], Lin-Feng Wu [2,\*], Zhong-Bo He [1], Song Peng [2] and Yun-Fei Fan [2]**

1. CNNC Key Laboratory of Uranium Resource Exploration and Evaluation Technology, Beijing Research Institute of Uranium Geology, Beijing 100029, China; wuyu@briug.cn (Y.W.); hezhongbo@briug.cn (Z.-B.H.)
2. Non-Ferrous Metals and Nuclear Industry Geological Exploration Bureau of Guizhou, Guiyang 550005, China; gzkywq@163.com (Q.W.); gzyshdypscc@163.com (S.P.); brainvan2008@163.com (Y.-F.F.)
* Correspondence: qiulinfei@briug.cn (L.-F.Q.); wulinfeng2022@126.com (L.-F.W.); Tel.: +86-010-6492-8279 (L.-F.Q.)

**Abstract:** Research on topics such as geological–tectonic evolution, metallogenic models of deposits (gold, mercury, lead, zinc, etc.), and ore-forming fluids' evolution has been conducted in Guizhou. However, few studies have been conducted on uranium (U) deposits (especially carbonate-hosted U deposits). Moreover, the relationship between hydrocarbon fluids and U-mineralization has not been addressed at all. Typical carbonate-hosted U deposits (including some ore spots) in Guizhou Province have been investigated through close field work, petrography, mineralogical, micro-spectroscopy, organic geochemical and C isotope studies. The central part of the U-ore body is often black (the black alternation zone) at the outcrop, and its sides are gray and gray-brown (the gray alternation zone); the color gradually becomes lighter (black to gray) from the center of the ore body out to the sides. Petrographic observations, microscopic laser Raman spectroscopy, and infrared spectroscopic and scanning electron microscope analyses have indicated that U-minerals (pitchblende and coffinite), pyrite and "black" organic matter (OM) are closely co-dependent, with the OM having the typical characteristics of bitumen. Large light oil fluid inclusions were found in gray alternation rocks (besides the U-ore body) with strong light blue fluorescence properties, indicating that hydrocarbon fluids and U-minerals may came from the same U-bearing hydrocarbon fluids. The values of the $^{13}C$ isotope value, a biomarker of OM and trace elements, REEs in U-ores, were found to be similar here to those in the local paleo-petroleum reservoir, indicating that the bitumen may originate from the deeply intruding paleo-petroleum reservoir. The precipitation of U is related to the cracking differentiation of hydrocarbon fluids. As result, the carbonate-hosted U-mineralization in Guizhou is neither of a sedimentary diagenesis type, nor of a sedimentary diagenesis superimposed leaching hydrothermal transformation type, as have been described by previous scholars. To be exact, the U deposit is controlled by fault and hydrocarbon fluids, and so it can be defined as a structural hydrocarbon–carbonate-type U deposit. A new U-mineralization model was proposed in this study. Here, U, molybdenum, and other metals were mainly found in the black rocks in the lower stratum (presumably Niutitang Formation), having migrated together with hydrocarbon fluids in the form of tiny mineral inclusions. The hydrocarbon fluids (containing some brine) caused cracking and differentiation upon entering the fracture zone, at which point the ore-forming materials (U, pyrite, and other metals) were released and precipitated.

**Keywords:** hydrocarbon fluids; cracking and differentiation process; carbonate-hosted uranium deposit; Guizhou



## 1. Introduction

Carbonate-type U deposits are a separate class in the IAEA's deposit classification, and they account for about 1.4% of the deposits that have been discovered to date [1]. In China,

carbonate-type U deposits are generally classified as a subclass of carbonaceous–siliceous–pelite-type deposits. The carbonaceous–siliceous–pelite type U deposit was one of the earliest industrial U deposits discovered in China in the 1950s, the demonstrated reserves of carbonaceous–siliceous–pelite type U deposits were accounted for 16% of the total U resources [1]. These deposits are small to medium-sized, widespread and have low to medium grades (<0.1–0.2% U), but the potential U resources are huge. Deposits of this type occur in the southern and central regions of China (Guizhou, Guangxi, Hunan, Jiangxi and Sichuan Province) [2]. Due to the cost of mining, the exploration of carbonaceous–siliceous–pelite-type deposits have effectively stopped. The carbonaceous–siliceous–pelite-type U deposits hosted in the marine carbonate rocks, which are mainly distributed on the margins of Paleozoic landmasses in China. In Guizhou, two types of carbonaceous–siliceous–pelite-type deposits are found in carbonate rocks and mudstones, The deposits in mudstone are formed by marine sedimentation, and the scale is large, ranging from large to super-large scale (such as the Longwan Deposit in southeast Guizhou), but there is no economic development value yet. The U deposits hosted in carbonate rocks, though smaller in scale than the hosted in mudstone at present, are of economic exploitation value. In fact, due to the lack of knowledge on the genesis of carbonate-hosted U deposits, these deposits are lack of attention due to simply classified as carbonaceous–siliceous–pelite-type U deposits. In Guizhou, there are a group of carbonate-hosted U deposits of a certain scale that includes the Baimadong deposit (medium-sized deposit), the Dayutang deposit (small-sized deposit), the Dajishan deposit (small-sized deposit), and hundreds of other U ore spots. To date, the Baimadong deposit is the only one to have been industrially mined.

Guizhou is famous for its world-famous Carlin-type gold deposits; our predecessors have studied its structural evolution, ore-forming fluid content, and its metallogenic model [3–5], and some scholars have discussed the metallogenic relationship between paleo-petroleum reservoirs and Carlin-type gold deposits [6,7]. Only a relatively small number of scholars have studied uranium deposits for their basic geological characteristics, such as their mineral composition [8–10]. In particular, the metallogenic genesis of carbonate-type uranium deposits remains unclear. Just after the discovery of uranium deposits (mineralization), scholars generally identified the carbonate-type U deposit in Guizhou as the sedimentary diagenesis type. Therefore, it has been classified as a carbonaceous–siliceous–pelite-type deposit. Since then, some scholars have argued that hydrothermal superimposition might be taking place during mineralization, as well as sedimentary diagenesis [11]. Indeed, several paleo-petroleum reservoirs have been discovered in Guizhou, including Majiang, Banjie, Anran, Baiqi, etc. These paleo-petroleum reservoirs are often associated with Carlin-type gold deposits, such as Lannigou, Shuiyindong, Banqi, Yata, etc. [12], and the mineralization of Carlin-type gold deposits may closely depend on hydrocarbon fluids. In fact, many scholars have focused on the genetic relationship between U-mineralization and hydrocarbon fluids in sandstone-type U deposits [13–18]. To date, no scholars have discussed the relationship between U-mineralization and hydrocarbon fluid in this type uranium deposit in Guizhou.

After the field geological survey, typical samples (including U-ores, alternation rocks and source rocks) were collected from the carbonate-hosted U deposits (or ore spots) in southwestern and northern Guizhou, and petrographic, mineralogy, fluid inclusion, trace Earth Elements (including REEs), C isotope and biomarker compound were analyzed by optical microscopy, fluorescence microscopy, scanning electron microscopy with X-ray energy-dispersive spectrometer (SEM-EDX), microlaser Raman, etc. This paper focuses on the relevant metallogenic mechanisms within, and the genetic relationship between, hydrocarbon fluids and the U-mineralization of carbonate-type U deposits in Guizhou.

## 2. Geological Setting

### 2.1. Tectonic Background

The study aeras located in the north and southwest of Guizhou Province. Guizhou belongs to the "Yangtze Continent Block" of the "Qiangtang–Yangtze–South China Plate",

which is divided by the tectonic regional and belongs to the passive marginal fold–thrust belt in the southern part of the Yangtze Block. It is composed of the Yangtze Continent Block, the Jiangnan Orogenic Belt and the Youjiang Orogenic Belt (Figure 1a,b) [19]. The U metallogenic belt is part of the coastal Pacific uranium metallogenic region, the Yangtze Continent Block U metallogenic province, and the central Guizhou–northwestern Hunan U metallogenic belt (Figure 1c) [20].

During its tectonic evolution, Guizhou has experienced a series of magmatic–tectonic–metamorphic deformation events, occurring during the Mesoproterozoic Jinning Period, the early Paleozoic Caledonian Period, the late Paleozoic Variocian Period, the Triassic Indosinian Period, and the Jurassic–Cretaceous Yanshan Period. The sedimentary structure of Guizhou was stereotyped in the late Indosinian Period, and the main strata, which developed on the southern consolidated basement, were established in the late Paleozoic–Mesozoic sedimentary. The regional uplift caused by the Indosinian movement caused the ocean to recede, thus finally giving shape to the current basin–platform sedimentary structure [21].

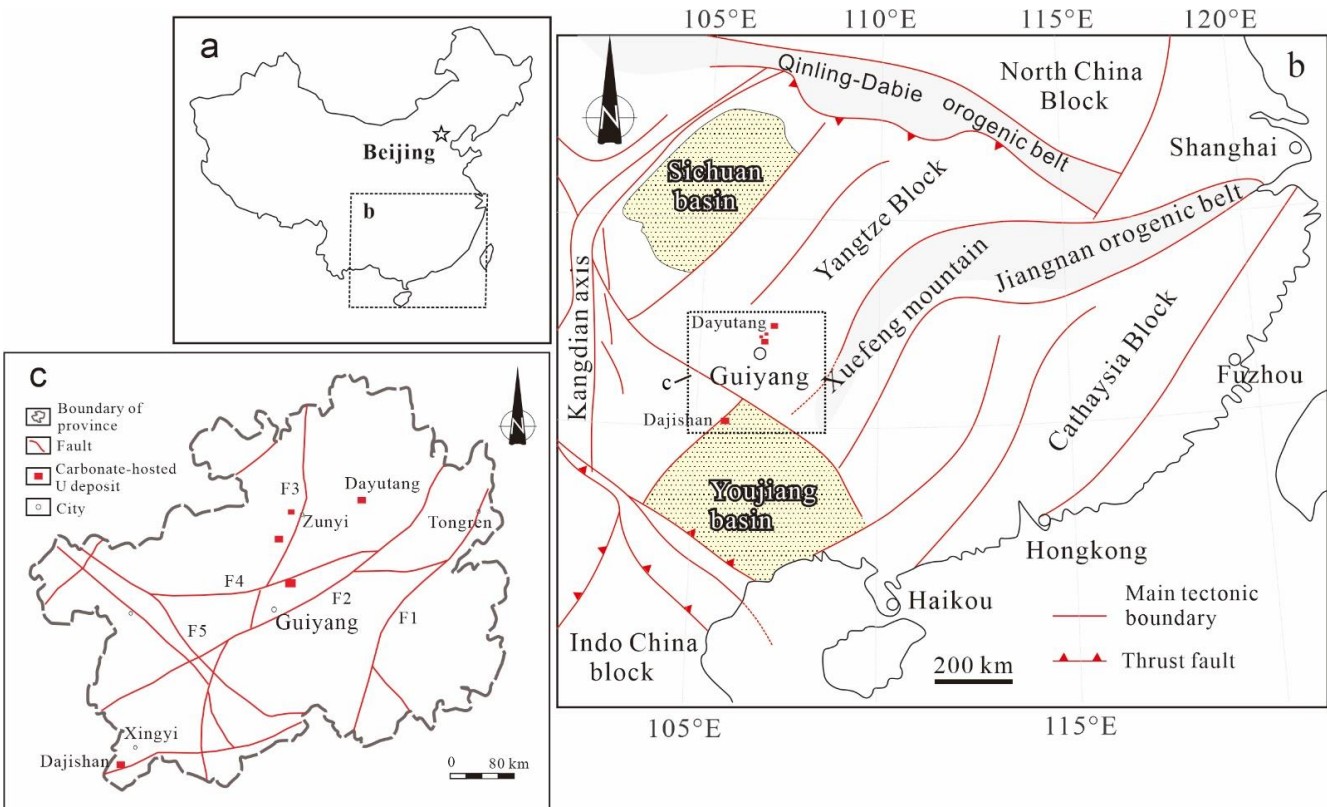

**Figure 1.** (**a**) The location of the study area shown in (**b**); (**b**) tectonic units and reginal of the study area, modified from [19,22]; (**c**) distribution of major carbonate-type uranium deposits in Guizhou Province.

### 2.2. Regional Stratigraphy and Faults

Regional stratas were exposed from the Mesoproterozoic to the Quaternary in Guizhou, and these are mainly characterized by the development of marine sedimentary rocks and an abundance of paleontological fossils. There are also abundant sedimentary minerals, such as coal, phosphorus, aluminum, and manganese. The Mesoproterozoic and Neoproterozoic sediments are dominated by marine terrigenous clastic rocks, mixed with pyroclastic rocks and carbonate rocks; the Paleozoic to the middle of the late Triassic sediments are composed of marine carbonate rocks mixed with clastic rocks, and the continental sediments were laid after the late Triassic. Magmatic rocks are rarely distributed in Guizhou, and some areas are exposed to basic dykes. The strata closely related to U-mineralization include the

Shiniulan Formation (Fm) (S$_2$*sh*), the Longtan Fm (P$_3$*l*), the Qixia Fm (P$_2$*q*) and the Maokou Fm (P$_2$*m*) in northern Guizhou, and the Feixianguan Fm (T$_1$*f*) and the Jialingjiang Fm (T$_1$*j*) in southwestern Guizhou.

The regional deep faults in Guizhou can be categorized into the three groups of trending northeast (F1, F2, F3), trending near east–west (F4) and trending north–west (F5) (Figure 1c). These faults are generally more than 150 km long. The NE-trending and near-EW-trending deep faults determine the distributions of carbonate-type U deposits.

### 2.3. Carbonate-Type Uranium Deposits

Since the 1960s, three medium–small carbonate-type U deposits (Baimadong, Dayutang and Dajishan) and many carbonate-type U-ore spots have been discovered in Guizhou, estimated to contain in excess of ~5000 tons of U in total, at an average grade of 0.03–0.05% U [9]. Among these deposits, Baimadong is medium-sized, but its shallow ore bodies have been fully operated, and the mine is currently closed. Due to the shallow depth of exploration (<200 m), the Dayutang (in north Guizhou) and Dajishan (in the southwest of Guizhou) U deposits are only small. No sampling has been carried out at the Baimadong deposit as it was closed after the mining operation ended, and so this study focuses on the Dayutang and Dajishan U deposits. The basic geological characteristics of Dayutang and Dajishan are thus now discussed in detail.

The Dayutang U deposit is in the south of Fenggang County, Zunyi City, Guizhou Province (Figure 1c), located on the eastern side of the Nanxiechuan inversion slope in the southern part of the Yangtze Block. The exposed strata surrounding the deposit mainly include the lower Ordovician Honghuayuan Fm, the Middle Ordovician Baota Fm, the lower Silurian Longmaxi Fm, the Middle Silurian Shiniulan Fm, the Hanjiadian Fm, the Middle Permian Qixia Fm, the Maokou Fm, the lower Triassic Yelang Fm and the Shabaowan Fm. U-mineralization mainly occurs in the structural breccia at the bottom of the Longtan Fm and at the contact point between the Qixia Fm and the Maokou Fm [23]. The ore bodies here are mainly controlled by the trending of near north–south faults (The strike is 10°–15°) (Figure 2a,b). The U-ores are grayish black or dark gray, and mainly display a breccia structure. The cements of U-ores are mainly composed of disseminated pyrite, kaolinite, and organic matter (OM). The grade of U-ores is generally between 0.05% and 0.1%.

The Dajishan U deposit is in the southwest of Xingyi City, Guizhou Province (Figure 1c), located in the intersection of the Tethys and coastal Pacific tectonic domains. The strata exposed around the deposit are mainly composed of carbonate rocks and clastic rocks of the lower Triassic Jialingjiang Fm, which can be divided into three lithologic segments (Figure 2c). The first component of the Jialingjiang formation (T$_1$*j*$^1$) is primarily carbonate rocks, intercalated with siltstone and a small amount of dolomite, and this is the main ore-bearing strata of the Dajishan U deposit; the second component of the Jialingjiang formation (T$_1$*j*$^2$) is purplish red argillaceous siltstone mixed with yellowish brown mudstone and argillaceous siltstone; the lowest part, and the third member, of the Jialingjiang formation (T$_1$*j*$^3$) is dominated by carbonate rock intercalated with dolomite. Several NE–NEE trending faults (with a strike of 40°–50°) have developed in the Dajishan U deposit. The U-ore bodies here are controlled by faults, occurring in the NE–NEE high-angle normal fault zone and the secondary structural fracture zone (Figure 2d), or in the interlayer fracture zones present within different lithologic interfaces. The U-ores are grayish black or dark gray, primarily with a breccia structure. The cements of uranium ores are mainly composed of disseminated pyrite, kaolinite, and OM. The breccias of the U-ores are primarily angular to sub angular. The grades of the U-ores are generally between 0.05% and 0.2%.

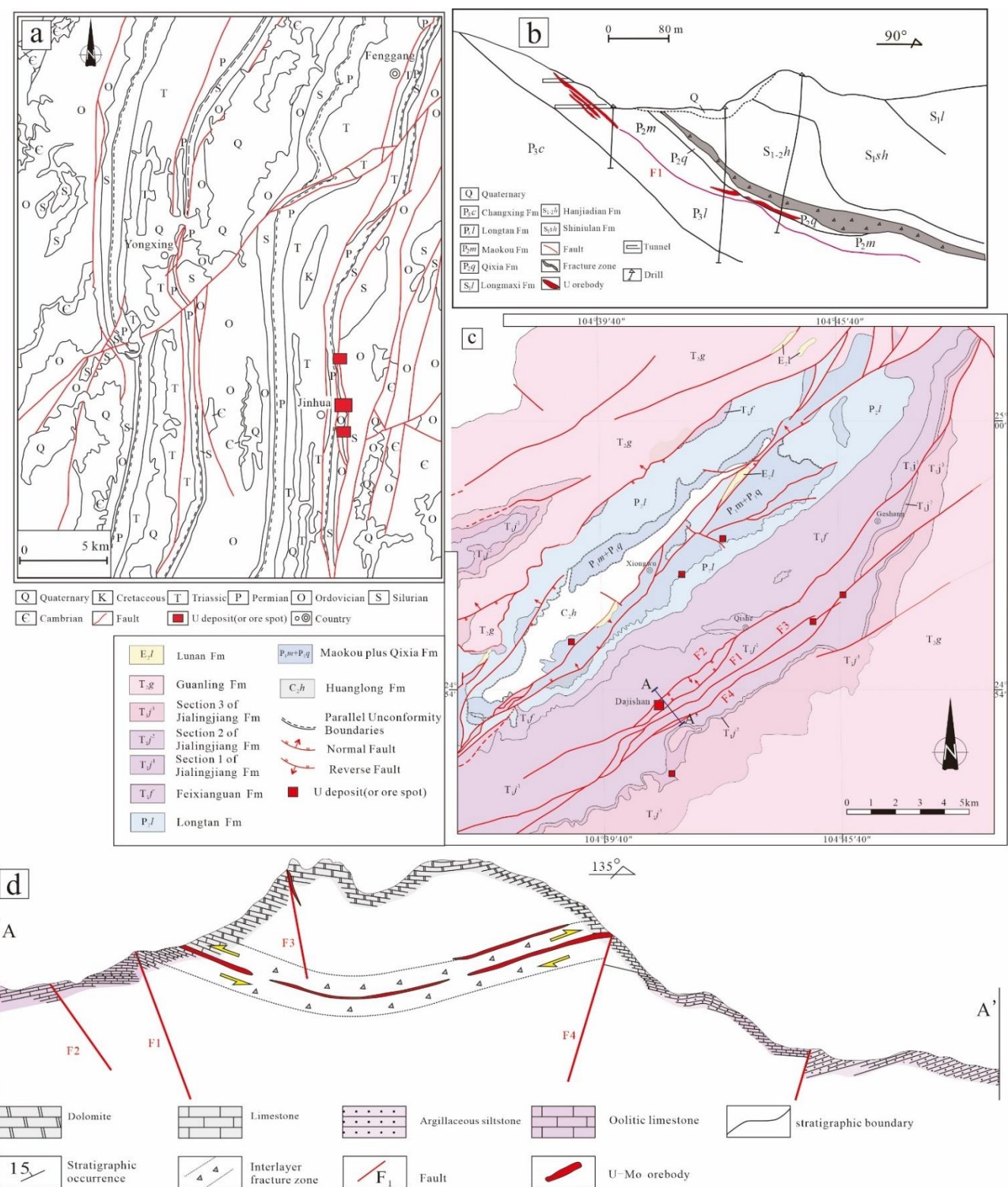

**Figure 2.** Geological map of the Dayutang U deposit (**a**) and a schematic cross-section showing the spatial positions of the U-ore bodies (**b**). Geological map of the Dajishan U-Mo deposit (**c**) and a schematic cross-section showing the spatial positions of the U-ore bodies (**d**), modified from [9,23].

## 3. Sampling, Analytical Procedures and Methods

### 3.1. Samples

More than 30 samples were used in this study, and these were collected from the old tunnels and outcrops of the Dayutang and Dajishan U deposits. Before laboratory studies, the samples were carefully observed and photographed. Mineralogical, fluid inclusion, quantitative REE, organic geochemical and isotope studies were carried out.

### 3.2. Mineralogical Study

The samples were subjected to mineralogical studies. There were 16 ore-bearing samples and 4 alternation zone samples, and microprobe and fluid inclusion sections were constructed. The mineralogical studies were conducted at the laboratory of the geology and mineral resources division of Beijing Research Institute of Uranium Geology.

#### 3.2.1. SEM and EDX Analysis

Back-scattered electron (BSE) images were obtained using a VEGA3 scanning electronic microscope (TESCAN, Brno, Czech Republic). The analytical conditions were 20 kV voltage acceleration. EDX analysis was conducted using an EDAX TEAM energy-dispersive spectrometer (AMETEK, Berwyn, PA, USA) with the following parameters: single point acquisition time 200 µs, input CPS more than 10,000, and dead time less than 30%.

#### 3.2.2. Fluid Inclusion Analysis

Fluid inclusion (FI) analyses were conducted on double-polished thin sections (200–300 µm thick). Petrographic observations were made using a Leica 4500 microscope. Microthermometry measurements (Leica MICROSYSTEMS, Wetzlar, Germany) were performed on the FI after calibration using a Leica 2500 microscope coupled with a Linkam THM600 heating–cooling stage (accuracy of $\pm 0.1$ °C). The freezing point temperature had an estimated precision of $\pm 0.1$ °C. The heating–freezing rate generally ranged from 1.0 to 5 °C/min, but decreased to <0.3 °C/min when approaching the transformation phase. The final ice melting temperatures were converted to weight percentage NaCl equivalent (wt. % NaCleqv.) using the equation of Bodnar (1994) [24]. The estimated accuracy of the homogenization temperature (Th) was $\pm 2$ °C.

#### 3.2.3. Raman Microprobe Spectroscopy

The thin sections were placed on the stage of an BX-41 microscope (Olympus, Tokyo, Japan) equipped with 10× to 100× objectives, which is part of the Evolution-type laser Raman microscope system (HORIBA, Tokyo, Japan); this system also included an electronically cooled CCD detector, an illuminant system, and a filter system. The Raman spectra were excited by a 532 nm YAG laser at a resolution of 1 $cm^{-1}$ with the following parameters: 100× objectives, a scanning range between 100 and 4000 $cm^{-1}$, a grating of 1800 gr/mm, a single point gaining speed of 8 s, and 4 accumulations. The spectrum was calibrated using 520.7 $cm^{-1}$ sections of a silicon wafer. Data processing and spectral manipulation, such as smoothing, peak analyzing and baseline correcting, were performed using the Labspec 6 of Horiba software.

### 3.3. Quantitative Analysis of Trace Elements and REEs

In total, 10 samples of U-ores and alternation rocks were collected for the quantitative analysis of the trace elements and REEs, and 2 samples of black mudstone from the Niutitang Fm were collected from northeastern Guizhou for comparison. After the samples were crushed to 80 mesh ($\Phi$ = 0.18 mm), trace elements and REE concentrations were measured using an ELEMENT XR Inductively Coupled Plasma Mass Spectrometer (ICP-MS) (Thermo Fisher, Waltham, MA, USA) via the whole rock dissolution method. The single-element detection limit was met at $10^{-9}$.

### 3.4. Organic Matter Extraction and GC-MS Analysis

The ore samples were crushed to 80 mesh ($\Phi$ = 0.18 mm), and extraction was completed with trichloromethane applied for 72 h, so as to collect the chloroform bitumen A (Bitm-A) content, with reference to the Soxhlet extraction procedure. The extracts were separated into saturated, aromatic, and polar (NSO) fractions. The saturated and aromatic hydrocarbon fractions were prepared for the next analysis.

GC-MS analysis was performed using a 5977A gas chromatograph–mass spectrometer (Agilent, Santa Clara, CA, USA). This system used an HP-5 fused silica capillary column (60 m × 0.25 mm × 0.25 μm i.d.). The GC oven was held isothermally at 120 °C for 2 min, then increased to 300 °C at the rate of 4 °C/min, and it then remained at this temperature for 12 min. The solvent was *n*-hexane, helium was the carrier gas, and the pressure was 15.475 psi. We used the standard substance perfluorotributylamine as a reference. The Bitm-A extraction experiment and GC-MS analysis were carried out in the State Key Laboratory of Oil and Gas Reservoir Geology and Exploitation, Chengdu University of Technology.

### 3.5. Isotopes Analysis

#### 3.5.1. Carbon Isotope of OM

The C isotope analysis of OM in the U-ores was performed on a system composed of Flash 2000 element analyzer (Thermo Fisher, Waltham, MA, USA) and a Delta v Plus gas isotope mass spectrometer (Thermo Fisher, Waltham, MA, USA). We prepared the OM samples for extraction by drying the U-ores in an oven for 24 h at 70 °C; they were wrapped in a tin cup that would be converted into $CO_2$ in the Flash 2000 Element analyzer (Thermo Fisher, Waltham, MA, USA ). Then, the $CO_2$ was inserted into the gas isotope mass spectrometer to test the carbon isotope composition via the application of helium through an online pipeline. The column used was an SS fused silica capillary column (3 m × 6 mm × 5 mm i.d.). The oxygen-passing reaction took 3 s, and the temperature of the reaction furnace was 960 °C. Helium was the carrier gas, and $CO_2$ was the reference gas (GBW04407, GBW04408 and IAEA-600). The standard deviation of the EA-GIMS for each measurement was less than 0.2%.

#### 3.5.2. Sulfur Isotope of Pyrite

An in-situ S isotope analysis was performed on a femtosecond laser ablation multi-receiver cup inductively coupled plasma mass spectrometer (fs-LA-MC-ICP-MS, Thermo Fisher, Waltham, MA, USA). Helium was used as the carrier gas. We set the laser conditions of line mode, 20 μm × 40 μm beam spot and a low frequency (6Hz) to ensure a stable signal would be obtained. The laser energy density was fixed at 1.02 J/cm². The sulfur isotope mass fractionation was corrected via the Standard-Sample-Bracketing (SSB) method, using pyrite reference substance Balmat; the matrix effect less than 0.5‰, and the result for the laboratory-standard pyrite JX was $\delta^{34}S$ = 16.8 ± 0.37 (2SD, N = 21). The experiment was completed at the National Geological Experiment Testing Center of the China Geological Survey. The analysis method is explained in detail by Qiu et al. (2021) [25].

All the above experiments were carried out at the Beijing Research Institute of Uranium Geology, except as specifically stated in the article.

## 4. Results

### 4.1. Petrography of OM and U-Minerals

#### 4.1.1. Petrography of Organic Matter

Under macroscopic inspection, the central part of a U-ore body is black or dark grayish, and the sides are generally gray or grayish brown (Figure 3a–c). The OM is found in veins disseminated in the middle fracture zone, between dolomite and argillaceous siltstone (Figure 3b,d), which is consistent with the distribution pattern of U-ore bodies. U-ores are black and dark grayish due to the abundance of OM (Figure 3e). Microcosmically speaking, OM is mostly found in the fractures, dissolved pores and open spaces of dolomite, argillaceous siltstone, and breccia-structure rocks (Figure 4a,b). OM present a spherical,

flowing structure, showing mobility characteristics. OM appears black under transmitted light, and fluoresces dark brown or not at all under UV light (Figure 4a–d). Strong blue fluorescence can be seen around the edges of OM, indicating the presence of light oil (Figure 4c,d). OM are in close symbiosis with U-minerals (pitchblende or coffinite) and pyrite. In the pores of argillaceous siltstone (alteration zone near to the ore body), strong light blue fluorescence also occurs under UV light (Figure 4e), indicating the presence of light oil charging events within U-ores; hydrocarbon fluids give rise to cracking and differentiation from the center of the ore body out to its sides. Many bitumen and light oil fluid inclusions (FIs), smaller than 1 μm to tens of microns in diameter, can be observed in the dissolution pores and fractures of limestone (Figure 4f,g). These FIs are mostly spherical or droplet-shaped, and gradually reduce in number towards the edges. These bitumen inclusions are found in the same places as hydrocarbon fluid inclusions.

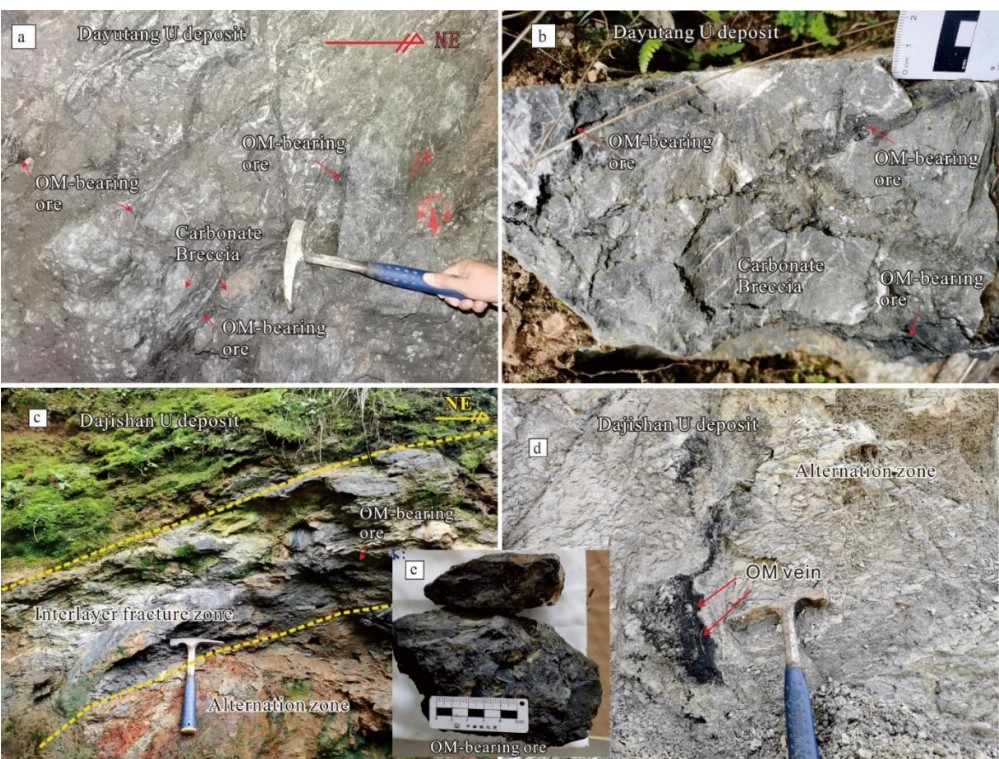

**Figure 3.** Photos showing the macroscopic features of OM and U-ore bodies in the Dayutang and Dajishan U deposits. (**a**,**b**) OM-bearing U-ores filling the fractures (around limestone breccia) in limestone in the Dayutang U deposit; (**c**,**d**) OM-bearing U-ores filling the normal and interlayer fractures in limestone (or argillaceous siltstone) in the Dajishan U deposit; (**e**) typical specimen of OM-rich black uranium ore. OM = organic matter.

### 4.1.2. Petrography of Uranium Minerals

There are two main types of U-minerals that can be determined by SEM-EDS analysis: pitchblende and coffinite. These U-minerals are both symbiotic with OM, pyrite and sulfur molybdenum and sphalerite etc. (Figure 5). U-minerals occur in three forms. In the first, they are scattered in pyrite, quartz, and sulfur molybdenum (Figure 5a–c,f); here, the particle size is generally between 1 and 2 microns or invisible as a mixture. The surfaces of the particles are smooth, and no dissolution holes appear under the scanning electron microscopy (SEM), which effectively excludes the possibility of U-minerals filling the fractures or dissolution holes. The presence of OM containing tiny pyrites and U-minerals indicates that the pyrite, U-minerals and quartz were formed at the same time. In particular, one can note the circle of OM (black under BSE) around the star-shaped U-minerals present in crystalline quartz and pyrite (Figure 5a,b,f). The second form of U-mineral emergence is

as irregular clumps in the interstitial positions of different mineral particles. The particle sizes of the blocky pitchblende are mainly 5–8 μm, but some reach 20 μm, and these mainly fill the intergranular pores of argillaceous siltstone or limestone (Figure 5d,h,i). The third form is as irregular veins, or filling reticulated fractures around carbonate breccias; in this form, the distribution is uneven. The veins are between 100 and 200 μm in width, and lots of pyrite and OM can often be found in the veins of U. These phenomenon shows that pitchblende (or coffinite) is close-symbiotic with pyrite, OM (pyrobitumen), sphalerite and sulfur molybdenum (Figure 5e,g,h,j).

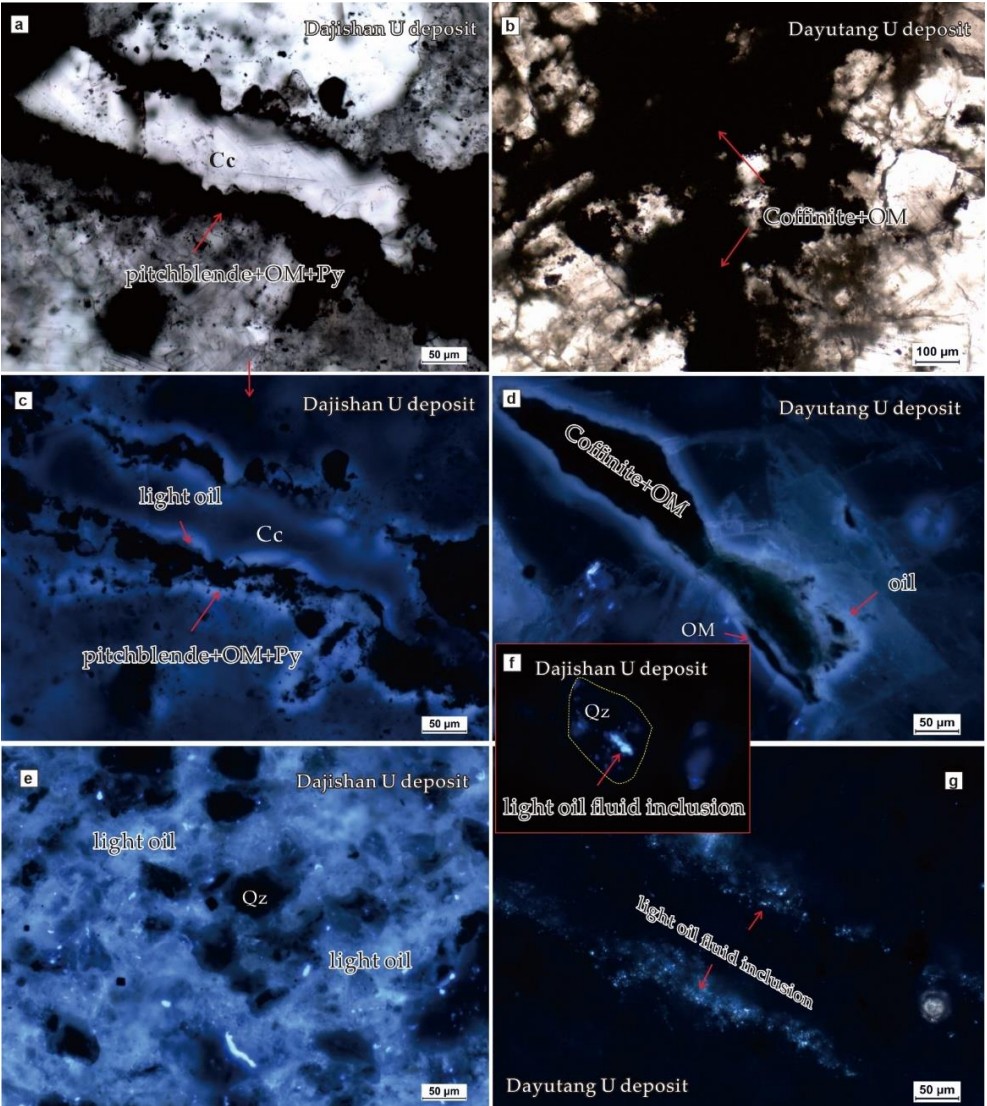

**Figure 4.** Pictures showing the microscopic characteristics of OM-bearing U-ore at the Dayutang and Dajishan U deposits. (**a**,**b**) Transmitted light. Uranium minerals and bitumen fill in the fractures (in limestone or argillaceous siltstone); (**c**) UV-stimulated fluorescence. The same area as photo "a"; here, bitumen does not fluoresce and exists in symbiosis with pitchblende, pyrite, and light oil, with light blue fluorescence evident around the edge of the bitumen in the Dajishan U deposit; (**d**) UV-stimulated fluorescence. Bitumen has no fluorescence and is not symbiotic with coffinite; light oil showing light blue fluorescence is evident around the edge of the bitumen in the Dayutang U deposit; (**e**) The argillaceous siltstone in the grey alteration zone around the ore body shows strong blue fluorescence in the Dajishan U deposit; (**f**,**g**) Numerous light oil and bitumen inclusions have been found in the fractures or pores of uranium ores in the Dayutang and Dajishan U deposits. Qz = quartz; Cc = calcite; Py = pyrite; OM = organic matter.

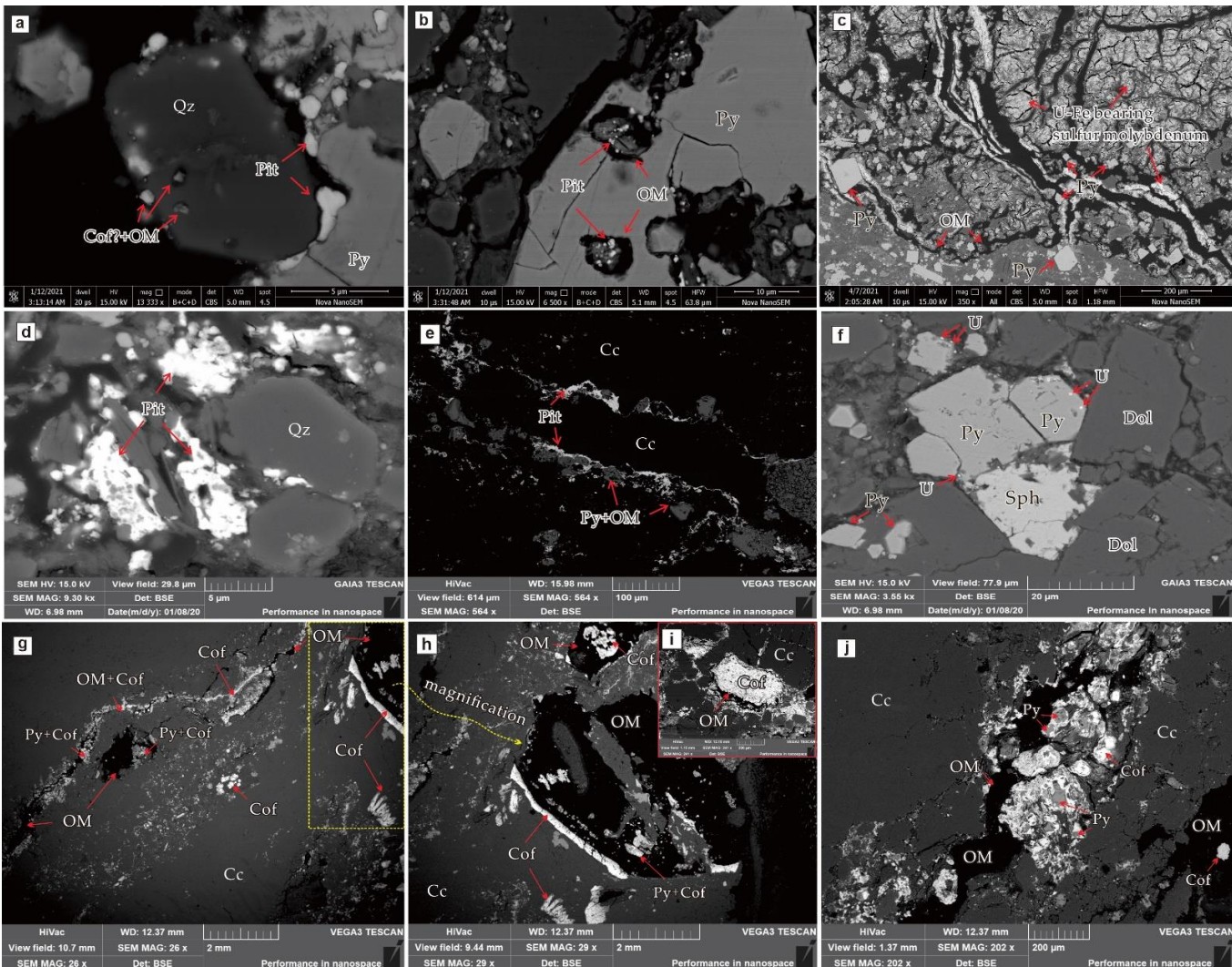

**Figure 5.** Pictures showing the distributions of U-minerals at the Dajishan (**a**–**f**) and Dayutang (**g**–**j**) U deposits. BSE pictures. (**a**) Pitchblende scattered in quartz. (**b**) Pitchblende co-exists with OM scattered in pyrite. (**c**) U-bearing sulfur molybdenum co-exists with OM and pyrite. (**d**) Blocky pitchblende filling the intergranular pores of argillaceous siltstone. (**d**,**e**) Irregularly veined and reticulated OM-baring U veins filling the fractures around breccias of carbonate, and (**e**) is the same area as Figure 4a. (**f**) Sphalerite is symbiotic with pyrite containing particularly tiny U-minerals. (**g**–**j**) Blocky coffinite, irregularly veined and reticulated OM-baring U veins filling the fractures of carbonate. U=uranium; Pit=pitchblende; Cof = coffinite; Qz = quartz; Cc = calcite; Py = pyrite; Dol = dolomite; Sph = sphalerite; OM = organic matter.

### 4.1.3. Energy-Dispersive Spectrometer (EDS) and Raman Analysis of OM

EDS and Raman analyses were carried out to assess the OM co-existing with uranium minerals. Raman analysis showed that it has two obvious peaks near to 1350 cm$^{-1}$ and 1580 cm$^{-1}$, with the Raman spectral characteristics (picks of carbon) of pyrobitumen (Figure 6c,d), and EDS analysis showed that this organic matter exhibits a strong carbon peak(Figure 6e,f).

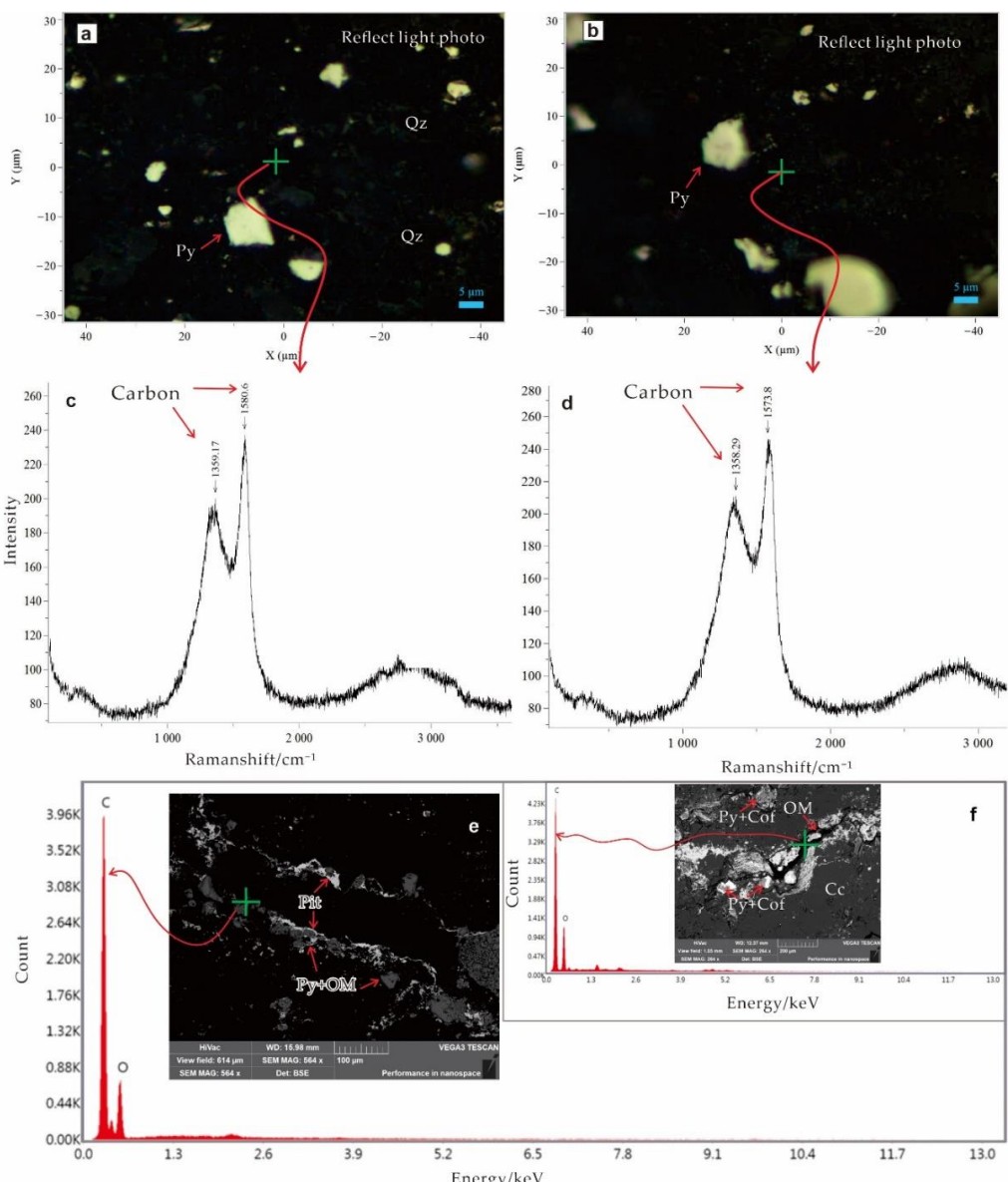

**Figure 6.** The Raman spectral (**c**,**d**) and EDS energy spectral (**e**,**f**) Characteristics of OM Co-existing with U-minerals in U-ores of carbonate-hosted U deposits, Guizhou. (**a**,**b**) Reflect light photos of the OM and pyrite in U-ore was shown the analysis location of Raman; (**c**) The Raman spectral characteristics of OM was collected in the image (**a**); (**d**) The Raman spectral characteristics of OM was collected in the image (**b**). (**e**) BSE image of the OM, pitchblende, pyrite was displayed in U-ore of Dajishan U deposit and the analysis location of the EDS. (**e**) BSE image of the OM, coffinite, pyrite was displayed in U-ore of Dayutang U deposit and the analysis location of the EDS. Qz = quartz; Py = pyrite; OM = organic matter; Pit = pitchblende; Cof = coffinite.

### 4.2. Fluid Inclusion

#### 4.2.1. Petrography of FIs

Petrographic studies show that the transparent minerals in siltstone-type uranium ores are mainly quartz minerals, with high roundness qualities and a diameter of 50–200 μm. Most of the quartz should be debris (Figure 7a,c,g), followed by a small amount of siliceous quartz with a very small particle diameter (Figure 7b). There are two main types of FIs in quartz debris: liquid-rich FIs and gas-rich FIs, the latter of which are distributed in quartz debris with a vapor proportion greater than 20%. These FIs have obvious origination characteristics and show no relationship with U-mineralization. Secondly, a small number

of hydrocarbon FIs, bitumen inclusions, liquid-rich FIs (vapor proportion < 5%) and pure gas FIs develop along the micro-fractures or in the dissolution holes of quartz, and pyrite is commonly found co-existing with these FIs (Figure 7a). Hydrocarbon FIs that fluoresce light blue are usually distributed along the micro-fractures of quartz debris (Figure 7c,d). These inclusions are related to U-mineralization, and indicate the presence of ore-forming fluids. A small proportion of U-mineralization is related to silicified quartz. This kind of quartz shows good crystal morphology, and is generally less than 20 microns in diameter. It is worth noting that some tiny globular pitchblende inclusions were found in the silicified quartz, and a black layer of OM was growing around the outside of the pitchblende (Figure 7b). This clearly indicates that pitchblende and OM have co-precipitation characteristics, meaning they were formed in the same stage.

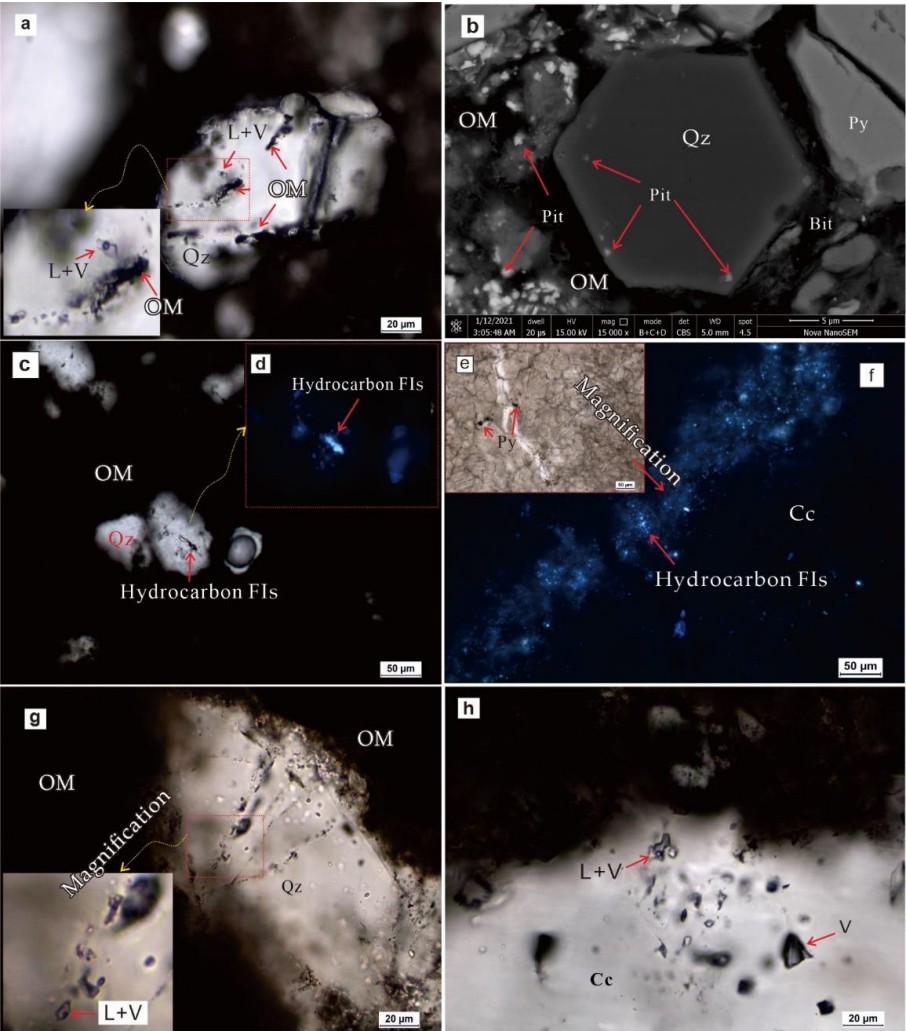

**Figure 7.** Microscopic pictures showing the FIs in the U-ores of carbonate-hosted U deposits, Guizhou. (**a**,**c**,**e**,**g**,**h**) Transmitted light image. (**b**) BSE image. (**d**,**f**) UV fluorescence image. (**a**) OM and liquid-rich FIs distributed along the micro-fractures in quartz debris of argillaceous siltstone U-ore. (**b**) Some tiny globular pitchblende inclusions scattered in the silicified quartz, and a black layer of OM growing around the outside of the pitchblende. (**c**,**d**) Hydrocarbon FIs fluorescing light blue distributed along the micro-fractures of quartz debris. (**e**,**f**) Lots of hydrocarbon FIs and some pyrite inclusions distributed through the micro-fractures of calcite with a breccia structure (carbonate) in the U-ore. (**h**) Liquid-rich FIs (vapor proportion < 10%) and pure gas FIs scattered in the cemented calcite of the carbonate-type U-ore. Pit = pitchblende; Qz = quartz; Cc = calcite; Py = pyrite; OM = organic matter. L = liquid; V = vapor.

Diagenetic-stage dolomite and calcite, and post-diagenetic stage calcite (or calcite cement), were identified in the dolomite and limestone via petrography. The FIs were recorded at the diagenetic stage and post-diagenetic stage (including the U-mineralization stage). These FIs are very complex and generally indistinguishable from one another, and will not be discussed in this article. We have identified calcite in the cemented state and in fractures closely associated with U-mineralization. Many hydrocarbon FIs, some liquid-rich FIs (vapor proportion < 10%), and some pure gas FIs were observed in the cementation and fracture-filling veined calcite. These FIs displayed the characteristics of ore-forming fluid (Figure 7e,f,h).

4.2.2. Microlaser Raman Analysis of FIs

After the petrographic analysis, FIs and mineral inclusions closely related to U-mineralization were selected for Raman analysis; 14 set of useful data were obtained. The analysis results are shown in Table 1. The laser Raman analysis of a single fluid inclusion shows that the gas component was mainly $CH_4$, and no other gas component was detected. The solid inclusions related to U-mineralization were mainly carbon (bitumen) and pyrite. The singular gas component ($CH_4$) and the number of solid carbonaceous inclusions (bitumen inclusions) in the U-ores indicate that these U-mineralization fluids are mainly related to the cracking of paleo-petroleum reservoirs.

**Table 1.** Composition of the inclusions in the U-ore detected by laser Raman from carbonate-type U deposits (or ore spots), Guizhou.

| No. Sample/Deposit | No. Test | Host Mineral | Distribution | Type of FIs | Raman Shift/cm$^{-1}$ | Compositon |
|---|---|---|---|---|---|---|
| DJS-1/Dajishan | 01 | Quartz debris | Along fractures | Solid inclusion | 1344, 1592 | Carbon |
| | 02 | Quartz debris | Along fractures | Solid inclusion | 1344, 1592 | Carbon |
| | 03 | Quartz debris | Along fractures | Pure gas FI | 2916 | $CH_4$ |
| | 04 | Quartz debris | Along fractures | Pure gas FI | 2911, 3020 | $CH_4$ |
| DJS-7/Dajishan | 01 | Quartz debris | Along fractures | Solid inclusion | 339, 376 | Pyrite |
| | 02 | Quartz debris | Along fractures | Solid inclusion | 1340, 1602 | Carbon |
| | 03 | Quartz debris | Along fractures | Solid inclusion | 1350, 1582 | Carbon |
| DJS-4/Dajishan | 01 | Cementation calcite | Clusters | Solid inclusion | 1307, 1558 | Carbon |
| | 02 | Cementation calcite | Clusters | Solid inclusion | 1350, 1602 | Carbon |
| | 03 | Cementation calcite | Clusters | Solid inclusion | 1322, 1547 | Carbon |
| | 04 | Cementation calcite | Clusters | Gas-liquid FI | 2916 | $CH_4$ |
| 102-1/Dayutang | 01 | Cementation calcite | Clusters | Pure gas FI | 2916 | $CH_4$ |
| 102-2/Dayutang | 01 | Veined calcite | Clusters | Solid inclusion | 339, 376 | Pyrite |
| | 02 | Cementation calcite | Clusters | Solid inclusion | 1322, 1547 | Carbon |

4.2.3. Homogenization Temperature (Th) and Salinity of FIs

The micro-thermometric analysis was conducted on the FIs related to U-mineralization, which were identified via a petrographic study. The micro-thermometric data of the Dajishan U deposit are listed in Table 2. For the Dayutang U deposit, no valid data were obtained, because the FIs were too small to be observed during the heating–cooling stage.

The Th of the FIs ranged from 70 °C to 172 °C, with an average of 128 °C. The Th histogram of Dajishan U deposit shows two obvious peak temperature ranges—70–90 °C and 130–160 °C (Figure 8a). The data in the low-temperature section (70–90 °C) mainly refer to the FIs that were symbiotic with the bitumen inclusions and those that showed no obvious symbiosis, and thus indicate the temperature of the early charging of hydrocarbon fluids, before the cracking of the paleo-petroleum reservoir. The data in the high-temperature section (130–160 °C) mainly refer to the FIs in symbiosis with OM, pyrite, and bitumen inclusions. OM is the representative substance of the U-mineralization stage, indicating that this is the temperature range of the stage at which hydrocarbon fluids begin to crack

and U-minerals precipitate. The peak temperature of the Dajishan U deposit should thus be in the range of 130–160 °C.

**Table 2.** Microscopic thermometry results of fluid inclusions in the Dajishan U deposit, Xingyi.

| Sample No./Lithology | Host Mineral | Distribution | Symbiosis | Number of FIs | Th/°C | Freezing Point/°C | Salinity/Wt%. |
|---|---|---|---|---|---|---|---|
| DJS-7-1/Argillaceous siltstone U-ore | Quartz debris | Along fractures | OM, Pyrite | 17 | 77–148 | −3.2−−6.7 | 5.26–10.11 |
| DJS-7-2/Argillaceous siltstone U-ore | Quartz debris | Along fractures | OM | 14 | 80–143 | −2.3−−7.0 | 3.87–10.49 |
| DJS-4-1/Broken Carbonate U-ore | Cementation calcite | Clusters | OM | 15 | 103–157 | −4.6−−10.2 | 7.45–14.15 |
| DJS-4-2/Broken Carbonate U-ore | Cementation calcite | Clusters | OM | 8 | 104–167 | −5.7−−6.8 | 8.81–10.24 |
| DJS-1-1/Argillaceous siltstone U-ore | Quartz debris | Clusters | OM, Pyrite | 3 | 132–137 | −4.5−−4.6 | 7.17–7.31 |
| | Quartz debris | Along fractures | OM, Pyrite | 15 | 72–171 | −4.5−−10.1 | 7.17–14.04 |
| DJS-1-2/Argillaceous siltstone U-ore | Quartz debris | Along fractures | OM, Pyrite | 12 | 97–145 | −5.0−−16.9 | 7.86–20.15 |

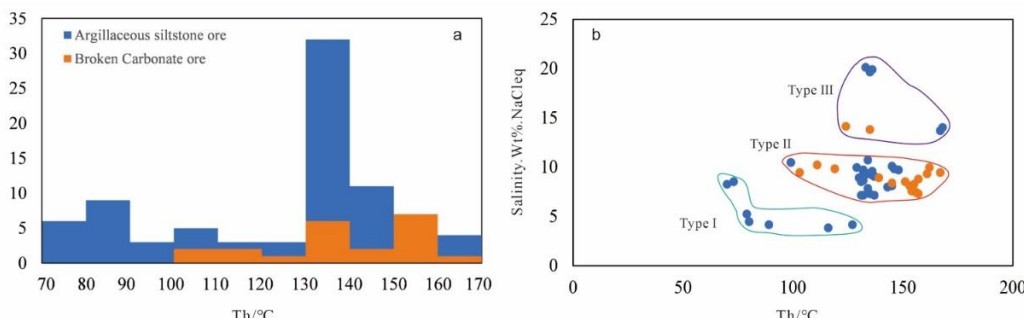

**Figure 8.** Homogenization temperature histogram (**a**) and its relationship with salinity dispersal in the U-ore of the Dajishan U deposit (**b**).

The freezing point ranges from −2.5 to −16.9 °C, and the corresponding salinity ranges from 4.18% to 20.15% NaCleq, indicating wide salinity variation (Table 2). According to the scatter point relationship between homogenization temperature and salinity (Figure 8b), the ore-forming fluids can be divided into three types: (1) low-temperature and low-salinity fluids, with temperatures between 70 °C and 120 °C and salinity levels less than 8% (type I); (2) low-temperature and low-salinity fluids (type II), with a temperature between 9 and 170 °C and a salinity level between 5% and 10%; (3) low-temperature and high-salinity fluids, with a temperature between 130 °C and 170 °C and a salinity level range of 15–20% (type III). These three types of fluids possess different genetic properties. Type I is probably primarily related to epigenetic low-salinity meteoric fluids, type III to deep hydrocarbon-rich high-salinity brine, and type II possess a mixture of the characteristics of type I and type III—it may be the product of a mixture of supergenetic low-salinity atmospheric precipitation and deep hydrocarbon-rich high-salinity brine.

### 4.3. Trace Elements in U-Ores, Alternation Rocks and Niutitang Fm Mudstone

25 trace elements (Li, Be, Sc, V, Cr, Co, Ni, Cu, Zn, Ga, Rb, Sr, Y, Mo, Cd, In, Sb, Cs, Ba, W, Re, Tl, Pb, Bi, Th, U) were tested from 11 samples (10 U-mineralization or alternation samples from the Dajishan deposit and Dayutang Deposit, 1 Mudstone from the Niutitang Fm), the results of 11 characteristic elements (V, Cr, Co, Ni, Cu, Zn, Mo, Cd, Re, Tl, U) of these elements, which are related to mineralization were shown in Table 3. The content of U, Mo, Re, Tl in U-ores is generally high (usually reaching more than 100 times, and with some elements more than 1000 times, the mean value of earth crust) in the U deposits, it is more obvious in the Dajishan deposit(Figure 9), Cr, Co, Zn, Cu are also enriched to a

certain extent. This is consistent with the large amount of pyrite (rich in Fe, Co, Ni), sulfur molybdenum (rich in Mo) and sphalerite (rich in Zn, Cu) in the U-ores observed by SEM. The contents characteristic elements (such as Mo, Re, U) in the mudstone of Niutitang Fm are similar to those of the U deposits, and the highest elements (Mo, Re, U) can also reach nearly 10 to 100 times the mean value of earth crust. Generally, the distribution curve of U-ore, the alteration zone samples, and the mudstone of the Niutitang Fm show consistent trends (Figure 9), the more intense the U-mineralization, the more concentrated U, Mo and Re.

**Table 3.** Characteristic trace elements of mudstone from Niutitang Fm, U-ores, and alternation samples for the Dajishan and Dayutang U deposits. The unit of concentration: $\times 10^{-6}$.

| Sample No. | Source | Sample Property | V | Cr | Co | Ni | Cu | Zn | Mo | Cd | Re | Tl | U |
|---|---|---|---|---|---|---|---|---|---|---|---|---|---|
| DJS-1 | | U-ore | 197 | 159 | 7.19 | 7.45 | 67.5 | 167 | 1004 | 2.6 | 0.062 | 1.97 | 615 |
| DJS-2 | | U-ore | 319 | 425 | 19.7 | 11.6 | 107 | 114 | 662 | 1.93 | 0.033 | 0.518 | 385 |
| PD-14-1 | Dajishan | U-ore | 418 | 442 | 81.6 | 440 | 109 | 208 | 11136 | 29.1 | 12.7 | 335 | 3724 |
| PD-14-2 | U-deposit | Alternation rock | 504 | 648 | 2.08 | 6.33 | 44.2 | 95.9 | 1240 | 0.319 | 0.009 | 0.326 | 8.45 |
| PD-14-3 | | Alternation rock | 386 | 550 | 11.4 | 37.1 | 78.1 | 84 | 348 | 1.33 | 0.169 | 11 | 22.2 |
| NTT-01 | Niutitang Fm | Mudstone | 4446 | 123 | 9.49 | 184 | 60.8 | 182 | 107 | 3.39 | 0.158 | 4.95 | 47.6 |
| 8610-1 | | Alternation rock | 34.8 | 3.37 | 30.9 | 46 | 0.874 | 6.19 | 2.9 | 0.03 | 0.005 | 0.208 | 18.8 |
| 8610-2 | | U-ore | 97.8 | 5.09 | 17.9 | 54.4 | 1.94 | 7.08 | 70.8 | 0.261 | 0.017 | 2.28 | 6242 |
| 8610-3 | Dayutang | U-ore | 58.1 | 6.47 | 22.9 | 50.5 | 2.48 | 5.29 | 5.83 | 0.055 | 0.003 | 0.69 | 735 |
| 102-1 | U-deposit | U-ore | 178 | 57 | 113 | 87.3 | 11.3 | 63.5 | 51.1 | 1.61 | 0.922 | 3.44 | 68.5 |
| 102-2 | | Alternation rock | 46.2 | 10.2 | 42.3 | 58.9 | 4.39 | 8.22 | 1.19 | 0.061 | 0.283 | 0.108 | 10.5 |

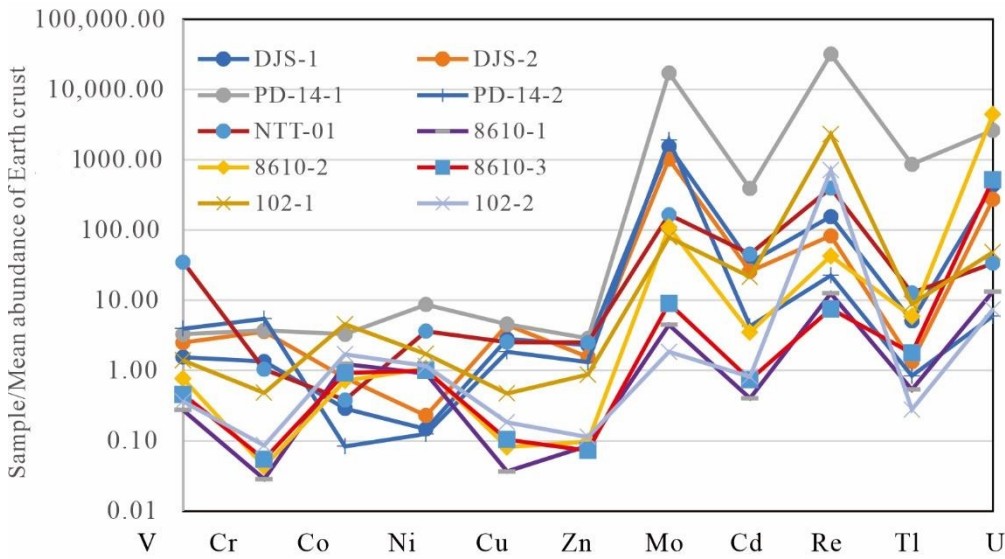

**Figure 9.** Characteristic trace elements change curves of mudstone from Niutitang Fm, U-ores and U-mineralization rocks from the carbonate-hosted U deposit.

### 4.4. REEs in U-ores,alternation Rocks and Niutitang Fm Mudstone

Key aspects of REE mineralization have been well documented ore-fluids and the process of mineralization [26,27], 7 mineralization samples and 2 source rocks samples were analyzed for comparative study, the testing data of REEs are listed in Table 3. The content of REEs in U-ores is generally high (usually reaching more than 10 times, and with some elements reaching nearly 100 times, the Clark value of chondrites) in the Dajishan U deposit (Figure 10). In the Dayutang U deposit, the content of REEs is slightly lower, but the sample of the alternation zone is significantly smaller. The RRE contents in the mudstone of Niutitang Fm are similar to those of the Dajishan U deposit, and the highest in the former can also reach nearly 100 times the Clark value of chondrites. Generally, the distribution curve of U-ore, the alteration zone samples, and the mudstone of the Niutitang Fm show consistent trends (Figure 10). They all dip towards the right, show similar LR/HR

ratios (La/Yb ratio), and exhibit clear negative δEu anomalies (Table 4), indicating that they may be from the same source.

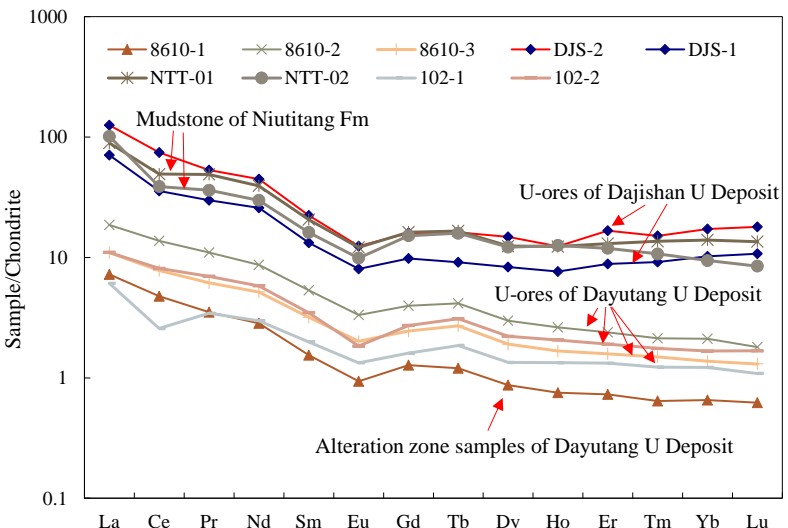

**Figure 10.** REE distribution curves of mudstone from Niutitang Fm and U-mineralization rocks from the carbonate-hosted U deposit.

**Table 4.** REE testing results of mudstone from Niutitang Fm, U-ores, and alternation samples for the Dajishan and Dayutang U deposits. The unit of concentration: $\times 10^{-6}$.

| Sample No. | La | Ce | Pr | Nd | Sm | Eu | Gd | Tb | Dy | Ho | Er | Tm | Yb | Lu | ∑REE | ∑LREE | ∑HREE | LR/HR | (La/Yb)$_N$ | δEu |
|---|---|---|---|---|---|---|---|---|---|---|---|---|---|---|---|---|---|---|---|---|
| DJS-1 | 22 | 28.8 | 3.64 | 15.5 | 2.58 | 0.591 | 2.54 | 0.433 | 2.68 | 0.549 | 1.86 | 0.314 | 2.14 | 0.346 | 83.973 | 75.651 | 8.32 | 9.09 | 6.93 | 0.23 |
| DJS-2 | 38.9 | 60.2 | 6.49 | 26.9 | 4.35 | 0.915 | 4.14 | 0.768 | 4.77 | 0.887 | 3.5 | 0.517 | 3.6 | 0.578 | 156.515 | 141.895 | 14.62 | 9.71 | 7.29 | 0.22 |
| NTT-01 | 27.7 | 39.8 | 5.97 | 23.6 | 4.03 | 0.893 | 4.2 | 0.787 | 4.01 | 0.886 | 2.74 | 0.466 | 2.92 | 0.436 | 118.438 | 106.193 | 12.25 | 8.67 | 6.40 | 0.22 |
| NTT-02 | 31.5 | 31.4 | 4.42 | 17.9 | 3.15 | 0.727 | 3.93 | 0.753 | 3.93 | 0.904 | 2.5 | 0.366 | 1.97 | 0.273 | 103.723 | 93.027 | 10.70 | 8.70 | 10.78 | 0.21 |
| 8610-1 | 2.24 | 3.85 | 0.428 | 1.7 | 0.301 | 0.069 | 0.33 | 0.057 | 0.28 | 0.054 | 0.153 | 0.022 | 0.137 | 0.02 | 9.641 | 8.918 | 0.72 | 12.33 | 11.02 | 0.22 |
| 8610-2 | 5.77 | 11.1 | 1.34 | 5.22 | 1.04 | 0.245 | 1.03 | 0.197 | 0.963 | 0.189 | 0.502 | 0.073 | 0.442 | 0.058 | 28.169 | 25.745 | 2.42 | 10.62 | 8.80 | 0.24 |
| 8610-3 | 3.4 | 6.29 | 0.751 | 3.1 | 0.622 | 0.148 | 0.633 | 0.128 | 0.614 | 0.12 | 0.333 | 0.051 | 0.287 | 0.042 | 16.519 | 14.944 | 1.58 | 9.49 | 7.99 | 0.24 |
| 102-1 | 1.89 | 2.08 | 0.42 | 1.79 | 0.388 | 0.098 | 0.415 | 0.088 | 0.432 | 0.096 | 0.278 | 0.042 | 0.255 | 0.035 | 8.307 | 7.081 | 1.23 | 5.78 | 5.00 | 0.24 |
| 102-2 | 3.42 | 6.56 | 0.851 | 3.49 | 0.678 | 0.135 | 0.705 | 0.147 | 0.712 | 0.148 | 0.4 | 0.06 | 0.349 | 0.054 | 17.709 | 15.839 | 1.87 | 8.47 | 6.61 | 0.20 |

### 4.5. GC-MS Analysis of OM in U-Ores

The GC-MS analysis results and the geochemical parameters of the extracts from the U-ores in the Dajishan U deposit are shown in Table 5. Most U-ores contain a certain amount of soluble OM, ranging from 0.0043% to 0.3615%. Here, the saturated hydrocarbon content is 20.0–44.12%, the aromatic hydrocarbon content is 3.53–7.14%, the colloid (nonhydrocarbon) content is 22.35–42.86%, and the bitumen content is 7.14–41.18% (Table 4). The Sa-HC/Ar-HC value is 5.7–7.5; the saturated hydrocarbon occupies the dominant position, reflecting the high maturity of the hydrocarbons in the U-ores, and the high quality of the oil-generating parent materials.

**Table 5.** Group composition and geochemical parameters of Soxhlet extracts from U-ores.

| Sample No. | Cont Bitm-A */% | Sa-HC/% | Ar-HC/% | Non-HC/% | Bt/% | Main C-peak | Sa/Ar | Pr/Ph | OEP | Pr/$n$C$_{17}$ | Ph/$n$C$_{18}$ |
|---|---|---|---|---|---|---|---|---|---|---|---|
| DJS-01 | 0.3615 | 44.12 | 5.88 | 38.24 | 11.76 | $n$C$_{18}$ | 7.5 | 0.765 | 0.878 | 0.272 | 0.386 |
| DJS-07 | 0.0966 | 42.86 | 7.14 | 42.86 | 7.14 | $n$C$_{17}$ | 6.0 | 0.853 | 1.076 | 0.300 | 0.382 |
| DJS-09 | 0.0043 | 20.00 | 3.53 | 22.35 | 41.18 | $n$C$_{16}$ | 5.7 | 0.931 | 1.02 | 0.512 | 1.927 |

\* Calculation formula of Content of Bt-A: $X = \frac{G2-G1}{m} \times 100\%$. $X$: Content of Bt-A, %; $G1$: Mass of empty weighing bottle (minimum value of two times), g; $G2$: Mass of weighing bottle plus Bt-A (minimum value of two times), g; $m$: Mass of sample, g. Sa: Saturated; Ar: Aromatic; Bt: Bitumen HC: hydrocarbon; Pr: pristane; Ph: phytane; OEP: odd-even predominace.

The carbon numbers of *n*-alkanes approximately range from $nC_{15}$ to $nC_{25}$, and they display a main C-peak of $nC_{16} \sim nC_{18}$ (Figure 11a). The low-molecular *n*-alkanes occupy the dominant position, showing that the OM is mainly sapropel-type, and the hydrocarbon source parent material contains fewer organisms (marine bacteria and algae). The Pr/Ph value is 0.765–0.9318 (Table 4), indicating that the depositional environment is anoxic with reducing conditions. The OEP (odd–even predominance) is 0.878–1.076, with no clear odd–even advantage, indicating that the hydrocarbon components of the U-ores are in the mature stage.

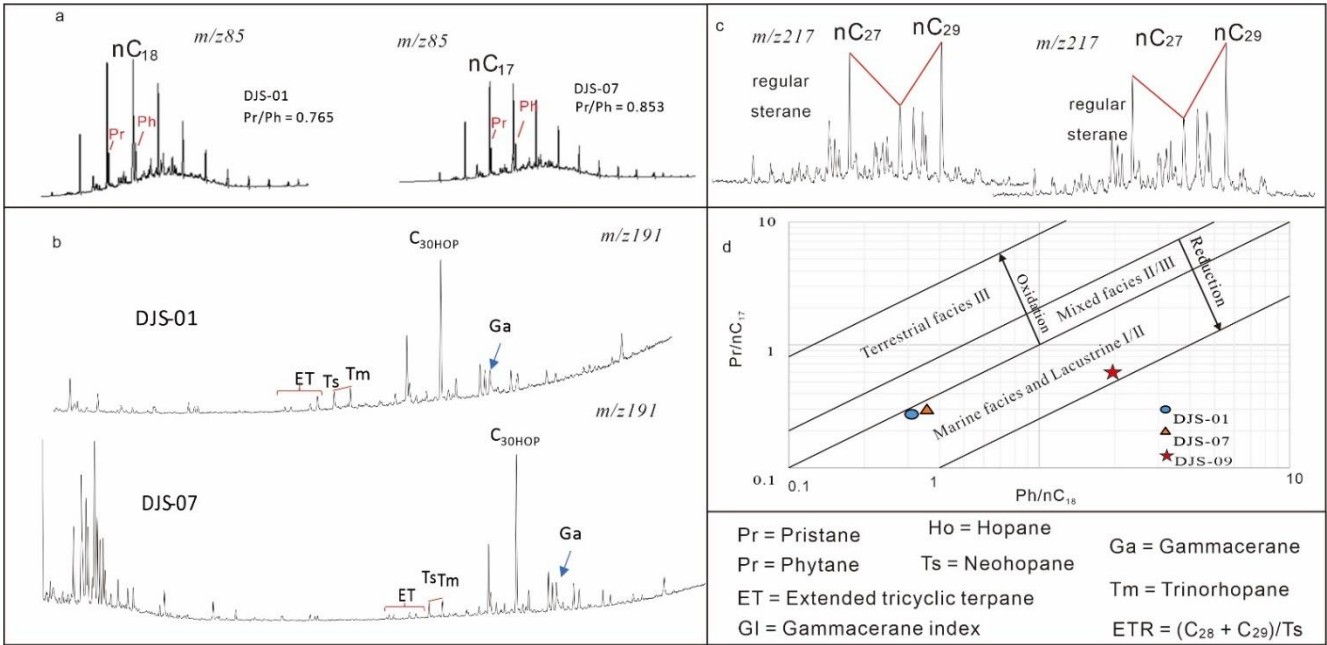

**Figure 11.** Gas chromatograms, ion mass chromatograms and $Pr/nC_{17}$–$Pr/nC_{18}$ plot of Soxhlet extracts of U-ores from the Dajishan U deposit. (**a**) Gas chromatograms of *n*-alkanes of U-ores; (**b**) characteristics of terpanes of Soxhlet extracts of U-ores; (**c**) characteristics of the steranes of the Soxhlet extracts of U-ores; (**d**) $Pr/nC_{17}$–$Pr/nC_{18}$ plot of Soxhlet extracts of U-ores. (**d**) $Pr/nC_{17}$ VS $Ph/nC_{18}$ plot of extracts in U-ores from the Dajishan U deposit.

The chromatogram of m/z 217 is dominated by pregnane and homopregnane. The $C_{27}$–$C_{28}$–$C_{29}$ regular steranes are distributed in the order of $C_{29} > C_{27} > C_{28}$ (Figure 11c). The pentacyclic terpanes are characterized by a high abundance of $C_{30}$ hopane, while the $C_{23}$ tricyclic terpane is predominant in the tricyclic terpanes (Figure 11b). This shows that the sedimentary environment was salty, giving algae greater advantages than bacteria. The Ts/Tm (Neohopane/Trinorhopane) value is 0.90–1.21, while the ETR (($C_{28} + C_{29}$)/Neohopane)) value is 1.88–2.23 and the GI (Gammacerane index) value is 0.36~0.39, indicating that the hydrocarbons formed in the reductive sedimentary environment of saline water. The $Pr/nC_{17}$ value is 0.272–0.512 and the $Ph/nC_{18}$ value is 0.386–1.927, and the $Pr/nC_{17}$–$Ph/nC_{18}$ correlation graph shows narrow distribution in the marine facies zone, and the OM belongs to type I or II (Figure 11d), indicating the formation of source rocks in a reductive environment.

### 4.6. Stable Isotopes

#### 4.6.1. C Isotope Analysis of OM from U-ores

The C isotopes of OM in the U-ore samples are shown in Table 6. The $\delta^{13}C$ isotope of the OM ranges from −28.8‰ to −27.2‰, with an average value of −28.0‰. This falls within the range of marine organic matter and sapropel kerogen ($\delta^{13}C$ of marine organic matter = −35‰–−10‰; $\delta^{13}C$ of sapropel kerogen = −30‰–−27‰) [28,29].

**Table 6.** $\delta^{13}C$ composition of Soxhlet extracts of ore-bearing sandstones.

| Sample No. | Deposit | Sample Properties | $\delta^{13}C$ V-PDB (‰) |
|---|---|---|---|
| DJS01 | Dajishan | Dark, argillaceous siltstone U-ore | −28.1 |
| DJS07 | Dajishan | Dark, argillaceous siltstone U-ore | −28.8 |
| DJS09 | Dajishan | Grey, broken Carbonate U-ore | −27.3 |
| 102-1 | Dayutang | Grey, broken Carbonate U-ore | −28.6 |
| 102-2 | Dayutang | Grey, broken Carbonate U-ore | −27.2 |

4.6.2. Sulfur Isotope of U-Symbiotic Pyrite

The sulfur isotope data of U-symbiotic pyrite from the U-ores are listed in Table 7. The $\delta^{34}S$ values of the pyrites co-existing with U-minerals are between −29.6‰ and −1.6‰ (mean of −17.5‰, N = 13) and are concentrated between −29.5‰ and −10.3‰, which are significantly negative values, these $\delta^{34}S$ values differ greatly from those of magmatic and metamorphic sources [30], and fall into the range of sedimentary organic matter [31]. It is worth highlighting that the $\delta^{34}S$ values of Dayutang U deposit are slightly higher than those of the Dajishan U deposit.

**Table 7.** $\delta^{34}S$ isotope compositions of Shazhou uranium in Xiangshan uranium ore field.

| No. Sample | Deposit | Mineral | $\delta^{34}S_{V-CTD}$(‰) |
|---|---|---|---|
| DJS-1-Py1 | Dajishan | Pyrite | −28.3 |
| DJS-1-Py1 | Dajishan | Pyrite | −28.5 |
| DJS-7-Py1 | Dajishan | Pyrite | −28.9 |
| DJS-9-Py1 | Dajishan | Pyrite | −29.5 |
| PD-14-Py1 | Dajishan | Pyrite | −22.9 |
| PD-14-Py2 | Dajishan | Pyrite | −22.2 |
| PD-14-Py3 | Dajishan | Pyrite | −14.3 |
| PD-14-Py4 | Dajishan | Pyrite | −1.6 |
| 8610-Py1 | Dayutang | Pyrite | −18.6 |
| 8610-Py2 | Dayutang | Pyrite | −10.3 |
| 8610-Py3 | Dayutang | Pyrite | −3.2 |
| 102-2-Py1 | Dayutang | Pyrite | −10.3 |
| 102-2-Py2 | Dayutang | Pyrite | −8.6 |

## 5. Discussion

### 5.1. Nature and Sources of OM

As mentioned above, black OM contributes significantly to U-mineralization; therefore, proving the nature and source of black OM is important when clarifying the metallogenic mechanism and establishing a metallogenic model. Generally, there are two types of OMs in sedimentary rocks: one is OM that entered and was preserved during the sedimentary process, such as plant detritus, peat, etc.; the other is OM sourced off-site, that is, brought in through fluid migration, such as of petroleum and organic acids, etc. [15]. The field geological survey showed that the black OM in the Dajishan and Dayutang U deposits is mainly veined or fine veined, and is disseminated in the interlayer fracture zone between dolomite (or limestone) and argillaceous siltstone. This is strongly controlled by the fault or interlayer fracture zone, and the OM is often produced together with the pyrite. The form in which the OM appeared here has clear characteristics of off-site migration.

Microscopically, the outer edge of the black OM was observed to strongly fluoresce light blue, indicating that the OM is cracked or differentiated. Laser Raman analysis showed

that the black OM displays two characteristic peaks near to 1350 cm$^{-1}$ and 1580 cm$^{-1}$, with peaks characteristic of typical carbonaceous substances. Stable isotope studies have shown that the $\delta^{13}$C value of the black OM in U-ore is −28.8‰ to −27.2‰ (average −28.0‰), which is closely comparable to that of marine OM and sapropel kerogen [28,29]. However, it is very different from humic kerogen (−26‰~−22.5‰) [25]. Sulfur isotopes of U-symbiotic pyrite show concentrations between −29.5‰ and −10.3‰, and these negative values suggest a typical sulfur-containing OM cracking type (TDS) or bacterial sulfate reduction (BSR) [31–33]. The above results suggest that the black OM in the U-ore of the Dajishan and Dayutang deposits is likely pyrobitumen formed by the cracking–differentiation of paleo-petroleum reservoirs, with obvious migration characteristics. The extracts of the U-ores contain abundant soluble OM (saturated, aromatic, and bitumen). The source material of hydrocarbons in the U-ores with high maturity and good oil-generating parents was provided by lower organisms (marine bacteria and algae). The geochemical parameters (Pr/Ph, OEP, ETR, order of regular steranes) all indicate the formation of source rocks in a reductive environment, and suggest that the hydrocarbons in the U-ores are in the mature stage.

Many paleo-petroleum reservoirs have been found in Guizhou Province. Previous scholars have carried out comparative studies on the organic geochemistry and the C isotope content between the paleo-petroleum reservoir and the Cambrian mudstone of the Niutitang Fm, and it is generally believed that the formation of paleo-petroleum reservoirs in Guizhou is related to the lower Cambrian Niutitang Fm. Chen's research showed that the $^{13}$C of kerogen ($\delta^{13}$Corg$\delta$) in the mudstone of the Niutitang Fm was about −31.51‰, and that the OM kerogen type was humus [34]. Yang also determined that the OM kerogen in the black rock of the Niuhetang Fm in Guizhou was humus-type [35]. Zhang's research showed that the $\delta^{13}$C values of crude oil and extracts from the Huzhuang 47 wells in the Kaili-Majiang region of Guizhou were −31.4‰ to −30.3‰; the $\delta^{13}$C values of bitumen in the Majiang paleo-petroleum reservoir were −29.9‰ to −29.3‰, while the kerogen in the black mudstone of the Niutitang Fm had humus-type characteristics, and its $\delta^{13}$C values ranged −31.9% to −29.1‰, with an average of −30.5‰. The organic geochemistry and C isotopic values of the paleo-petroleum reservoir are highly comparable with those of the lower Cambrian Niutitang Fm [36]. Li determined that the $\delta^{13}$Corg value of the black OM in the Baimadong U deposit was −29.7‰ to −25‰, suggesting that the OM is mainly derived from the Niutitang Fm [37]. Based on the form of the output, the microscopic features, the C isotope values, and the organic matter geochemical properties of the U-ores in the Dajishan and Dayutang deposits, it is concluded that the black OM here is mainly derived from paleo-petroleum reservoirs, and the source rocks are probably primarily from the lower Cambrian Niutitang Fm.

*5.2. Relationship between OM and U-Mineralization*

The physicochemical role of OM in U-mineralization is complex, and can be summarized as co-ordination, reduction and adsorption [38]. This study shows that the black OM is pyrobitumen, formed by the cracking of the paleo-petroleum reservoir; what role the paleo-petroleum reservoir plays in the U-mineralization process is yet to be determined.

Previous authors have generally stated that hydrocarbon fluids play two roles in U-mineralization: one is entering the surrounding rock before the U-mineralization, after which mineralization occurs to reduce the U- and oxygen-containing fluid; the other is to carry minerals and hydrocarbon fluids in their migration [39,40]. The ore body output of the carbonate-type deposit in Guizhou has the following important characteristics: (1) it is strictly controlled by the vertical and interlayer fracture zones, and has a layered and lens-like structure; (2) the center of the surrounding U-mineralized rock is dominated by OM and sulfide, and its color gradually becomes lighter from the U-mineralization center to the alternation zone. Furthermore, the U-ores generally have brecciated, disseminated and veinlet-like structures. Therefore, it is clear that the output of the ore body is strictly controlled by its structure and the black OM. Moreover, there is no obvious

development of tectonic hydrothermal veins visible to the naked eye, indicating that the U-mineralization fluid of the carbonate-type U deposit in Guizhou is not a common low-temperature hydrothermal U deposit, and the ore-forming fluid will be a hydrocarbon-rich, water-containing hydrothermal fluid.

The fluid inclusion study showed that the Th of the FIs in the Dajishan U deposit had two peak ranges of 70–90 °C and 130–160 °C. The data for the low-temperature section (70–90 °C) were mainly derived from the FIs that are symbiotic with the bitumen inclusions and a few brine inclusions, thus giving us the temperature of the hydrocarbon fluids' early filling stage, when the paleo-petroleum reservoir has not yet been cracked. The data for the high-temperature section (130–160 °C) were mainly derived from the brine FIs that are symbiotic with OM and pyrite (typical U-mineralization stage substance), and the Th of the symbiotic FIs tells us the limitation temperature at which the hydrocarbon fluid cracked and the U-minerals begin to precipitate. This shows that the peak mineralization temperature of the Dajishan U deposit should be above 130 °C (most likely between 130 °C and 160 °C; the cracking of paleo-petroleum reservoirs occurs at s similar temperature), and the salinity of the FIs indicates the mixing of low-salinity epigenetic atmospheric precipitation with deep high-salinity hydrocarbon-rich hydrothermal fluid. The compositions of the FIs are essentially $CH_4$ and carbonaceous; no other components were found, indicating that the ore-forming fluid is mainly composed of hydrocarbons.

The scanning electron microscopy studies also found that fine-grained bitumen U inclusions often exhibit a black carbonaceous membrane around the edges of the U-minerals in some silicification-genesis quartzes and pyrites, which indicates that U and bitumen co-migrate and co-precipitate. Trace elements and REE Quantitative Analysis indicated that U, Mo, Re, Ti, Tl, etc. and REE are enriched during U-mineralization, the variation trend of element content of U-ores are consistent with the black mudstone of Niutitang Fm. In the study area, except for the mudstone of Niutitang Fm, the stratas below the U deposits are generally depleted U (all stratas are less than $10 \times 10^{-6}$). It indicates that these enriched elements (U, Mo, Re, Ti, Tl, REE, etc.) may be derived from the strata of Niutitang Fm under the ore-body and may have migrated with the fluid. In addition, in the study area, neither the overlying formation nor the ore-bearing layer itself have undergone post-oxidation, which effectively excludes hydrocarbons as the agents that reduce the U and oxygen-containing fluids that carry the metallogenic materials and permit co-migration.

### 5.3. The Metallogenic Processes and Genesis of Carbonate-Types U Deposit

Our predecessors generally place the U deposits in carbonate rocks in southwest China into the category of "Carbonaceous–siliceous–pelite" deposits, which indicates a lack of understanding, and the belief that this is a type of U deposit based on sedimentation. China has not yet adopted the carbonate-type U deposit as an independent industrial classification, even though the International Atomic Energy Agency (IAEA) has listed carbonate-type U deposits as an important industrial type [41]. Recent studies have found that some carbonaceous–siliceous–pelite-type uranium ores endowed with normal uranium levels have the mineralization characteristics of typical hydrothermal U deposits from the south of China [37,42,43]; unlike the more common U deposits that are altered by hydrothermal superposition in U-rich formations, these are considered separate from hydrothermal U deposits [44].

The carbonaceous–siliceous–pelite U deposits in China are reclassified into three genesis types: the sedimentary–diagenetic sub-type, the exogenous infiltration sub-type and the hydrothermal sub-type [44]. Previous researchers stated that the genesis of the Dajishan U deposit adhered to the sedimentary diagenetic type or the sedimentary diagenetic superposition leaching–hydrothermal transformation type [8,9], while the genesis of the Dayutang U deposit is generally considered to be of the hydrothermal type [23]. In this paper, we have state that the developments of the Dajishan and Dayutang U deposits are similar, and are mainly controlled by faults and the cracking of paleo-petroleum reservoirs. U and other polymetallic metals may have migrated with paleo-petroleum fluids, in the

form of mineral inclusions with micro- and nanostructures. Hydrocarbon fluids mainly originate from the black rock system in the lower strata (presumably the Niutitang Fm), during which hydrocarbon fluids interact with stratigraphic brine, and this dissolves more mineralizing elements such as U, Mo, Ti, etc., to form nano-structured captives that enter the hydrocarbon-rich hydrothermal fluid and co-migrate. When the temperature and pressure change, the hydrocarbon-rich hydrothermal fluid cracks and differentiates. U, pyrite, and other metal substances, as well as heavy components (bitumen), are released and precipitated in the fracture zone, while light components (light oil, water) escape, but some are captured in calcite, quartz, etc., in the form of inclusions. The U-mineralization model and metallogenic process are shown in Figure 12. The possible chemical reaction formula of U-organic complexation and decomposition, described by Landais (1996) [45], is shown here:

$$2R\text{-}COOH + UO_2{}^{2+} \rightarrow RCOO\text{-}(UO_2)\text{-}OOCR + 2H^+ \tag{1}$$

$$2(RH) + UO_2{}^{2+} \rightarrow 2R^0 + 2H^+ + UO_2 \tag{2}$$

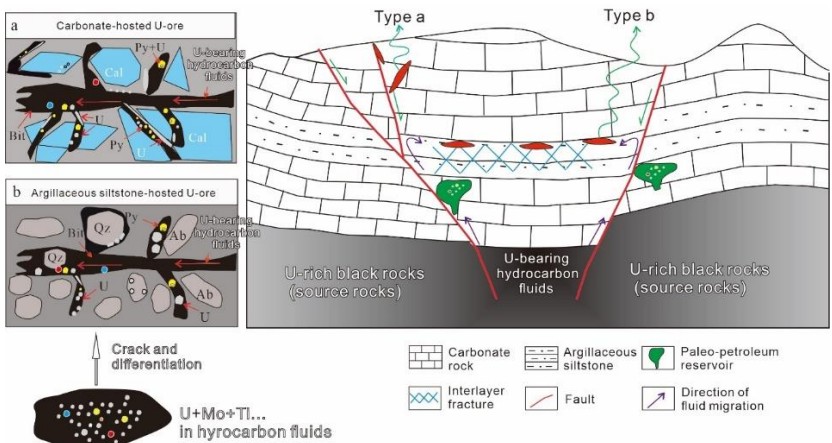

**Figure 12.** The conceptual model of U-mineralization and the metallogenic process of the typical carbonate-hosted U deposit in Guizhou. U = uranium minerals; Py = pyrite; Cal = calcite; Qz = quartz; Ab = feldspar. (**a**) Conceptual diagram of U-mineralization in carbonate rocks after cracking and differentiation of U-bearing hydrocarbon fluids; (**b**) Conceptual diagram of U mineralization in clastic rocks (Interlayers in carbonate rocks) after cracking and differentiation of U-bearing hydrocarbon fluids.

In fact, many have claimed that there is an important genetic connection between U-mineralization and paleo-reservoir cracking. In the carbonate formation of the Witwatersrand Basin in South Africa, Fuchs et al. (2017) found uranium–gold–lead and other polymetals in the form of nano-mineral inclusions, involved in the co-migration and precipitation of hydrocarbon fluids [14]. Therefore, the U-mineralization of the Dajishan and Dayutang carbonate-type U deposit may be neither the sedimentary diagenetic type previously declared, nor the superimposed leaching–hydrothermal transformation type of sedimentary diagenetic rock, but may instead be the structural–hydrocarbon carbonate type, controlled by the combination of faults and hydrocarbon fluid cracking.

## 6. Conclusions

Our research has identified the type of OM in the U-ore of the carbonate-hosted U deposit in Guizhou, and provided direct evidence that mobile hydrocarbon fluids participate in U-mineralization. We have proposed a new model of U-mineralization. In the new model, U and other metals can migrate together with hydrocarbon fluids in the form of mineral inclusions (with micro- to nanostructures), which may primarily come from the black rocks in the lower stratum (presumably the Niutitang Formation). During migration, the hydrocarbon fluids interact with brine, dissolving U and other metals in the formation of the ore-forming fluid. The microscopic mineral inclusions enter the hydrocarbon fluid

and migrate with it. With the change in temperature and pressure, the hydrocarbon fluid is cracked and differentiated, and the ore-forming materials are released and precipitated, resulting in U, pyrite, and other metal substances, while the heavy components (bitumen) remain in the fracture zone and form the U deposit.

**Author Contributions:** Conceptualization, L.-F.Q. and Y.W.; data curation, L.-F.Q., Y.W. and Z.-B.H.; formal analysis, L.-F.Q.; investigation, L.-F.Q.; methodology, L.-F.Q. and Z.-B.H.; project administration, L.-F.W. and Q.W.; resources, L.-F.W., Q.W., S.P. and Y.-F.F.; writing—original draft, L.-F.Q.; writing—review and editing, L.-F.Q. and Y.W. All authors have read and agreed to the published version of the manuscript.

**Funding:** This research was funded by National Natural Science Foundation of China, grant number U2167210, Geological Exploration Fund of Guizhou Province, grant number MCHC-ZG20212206-2 and Geological Exploration Fund of China National Uranium CO., Ltd., grant number 202133-2.

**Data Availability Statement:** Data sharing is not applicable to this article.

**Acknowledgments:** We are highly thankful to Geologic Party No.280, China National Nuclear Corporation for the field work investigation in the study area of Dajishan U deposit. Our thanks are also given to the editors and reviewers for constructive comments that helped improved the paper.

**Conflicts of Interest:** The authors declare no conflict of interest.

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
