# Peer review of "Metallogenic Mechanism of Typical Carbonate-Hosted Uranium Deposits in Guizhou (China)"

_minerals, doi:10.3390/min12050585_

Round 1

Reviewer 1 Report

This is very interested contribution to evolution of uranium ore deposits. However, I think that the representative analyses of uraninite and coffinite could be added and discussed.

The presented manuscript is very interested and hopeful for all researchers which are interested about uranium mineralisation. However, the carbonate-type uranium ore deposits form also very small group of mined uranium deposits, but the described deposits in Guizhou Province of China are very significant and very important for study of uranium mineralisation. For these deposits is highly interested relation of uranium mineralisation with organic matter. The detailed study and description of the uraniferous organic matter is one from very significant and important part of this manuscript. The other significant point of this paper is very nice and hopeful definition of hydrocarbon-carbonate-type uranium deposits.

The all presented data could by very useful for all researches, which are interested on study of uranium mineralisation on the world. However, I think, that could be for these scientists also very interested and hopeful presentation of some microprobe analysis of uraninite and coffinite, which form main part of this mineralisation.

Author Response

Dear revierwer;

  Thanks a lot for you hard work on my manuscript,The comments are very useful for me to improve the quality of my manuscript. I am agree that some microprobe analysis of U-minerals(pitchblende and coffinite) is useful for understanding the mineralization, but I believe that the EDAX anlysis (semi-quantitative) is enough for discriminate U-minerals,  the anlysis result of EDAX does not affect the conclusions of this study. Of course, when I do my next research, I will operate the analysis of microprobes。

Reviewer 2 Report

  • In title, the Carbonate-Type, also used as carbonate U-type in the manuscript, should be replaced by Carbonate-hosted.
  • In Fig. 1, why there are only 2 deposits in Fig. 1B.
  • Why the peak shift of the OM are so different in Fig 6A and 6B.
  • How did the authors know hydrocarbon fluids and U-minerals may come from the same U-bearing hydrocarbon fluids? What kind of U-bearing hydrocarbon fluids it is?
  • The authors claim that the precipitation of U is related to the cracking differentiation of hydrocarbon fluid. Please show more evidence.
  • Uranium, molybdenum, and other metals were … migrated together with hydrocarbon fluids in the form of tiny mineral inclusions. Please show more evidence about the same origin or various sources of polymetallic elements.
  • In the Discussion of the manuscript, Lines 620-628, it is said that the mineralization are controlled by the combination of faults and hydrocarbon fluid cracking. Did the ore-forming materials (U, pyrite, and other metals) were released and precipitated at the same site, as shown in the Lines 36-40?
  • In Geological Setting, you should stand a geological view, in place of a Guizhou province view. In the Introduction of the manuscript, Lines 46-60, the authors claim that these deposits of this type are distributed in the southern and central regions of China (Guizhou, Guangxi, Hunan, Jiangxi and Sichuan Province), however, there are too litter about these deposits of this type, not only in Geological Setting, but also in the Discussion of the
  • In Fig. 9, there are only 5 lines which has been marked, however, the others are still not been known well.
  • The English of the manuscript is to be improved.

Author Response

Dear reviewerï¼›

Thank you very much for the recommend of my manuscript,these suggestions are very helpful for improving my manuscript. I had revised the manuscript in this days,details as below:

  1. I added some U-deposit or ore spot In Fig. 1;

2.The Raman shift of the OM are generally a range(D peak is range of 1330~1380cm-1,G peak is range of 1560~1600cm-1), is related to the degree of order of carbon matter, it is normal for samples from different locations to have different degrees of order;anyhow, I changed the a Raman picture to make them more consistent;

3.The paper adds more microscopic evidence of mineralogy, evidence of trace element variation curves of uranium ore,alternation zone rocks and lower black uranium-rich mudstone(Niutitang Fm), all of which support that the ore-forming materials may originate from the same set of fluids ï¼›

  1. It is common phenomenon that the ore-forming materials (U, pyrite, and other metals) were released and precipitated at the same site(fault and interlayer interface)ï¼›

5.Other lines was supplementary marked in the Fig. 9.

6.The English of the manuscript has been edited by MDPI. Please see the Edit certificate.

  1. Other revisions are detailed in the manuscript.

Thanks again for your hard work on my manuscript.

Best regards,

l. f. Qiu 2022-4-19

Round 2

Reviewer 1 Report

The presented paper is very interested.

Author Response

Dear Reviewer,

 Thanks A lot.

Best regards

Qiu

Reviewer 2 Report

The authors have conducted a satisfy revision and I think it is ready for publication.

Author Response

Dear Reviewer,

Thanks again for your hard work.

Best regards,

Qiu   2022-4-24